# Gabapentinoids confer survival benefit in human glioblastoma

Joshua D. Bernstock [1,2,3,17] ✉, Mulki Mehari[4,17], Jakob V. E. Gerstl[1,5,17], David M. Meredith[6], Pablo A. Valdes[7], Philip Heesen[5], Vardhaan S. Ambati [4], Saritha Krishna[4], Jason A. Chen[1], Harshit Arora[1,5], Benjamin R. Johnston [1,2], Mikias Negussie [4], Kristina Grieco[6], Cesar Nava Gonzales[4], Abraham Dada [4], Anna B. Lebouille-Veldman[1,5], Aymen Kabir[4], Lennard Spanehl [1,8], Jasleen Kaur[4], Sena Oten[4], Youssef E. Sibih [4], Rania A. Mekary [5,9], Andy Daniel[4], John Frederick de Groot [4,10], Saef Izzy [11], Florian A. Gessler[8], Scellig Stone [2], Omar Arnaout[1], Yi Lu [1], Bryan D. Choi[12], Matthew D. Hall[13,14], Minesh P. Mehta [14], Yazmin Odia [13], Gregory K. Friedman [15], E. A. Chiocca [1], Pier Paolo Peruzzi [1,18], Timothy R. Smith[1,5,18] & Shawn L. Hervey-Jumper [16,18] ✉

Neuronal-glioma interactions are increasingly recognized as critical in the development and progression of central nervous system tumors. Recent research highlights that gliomas can integrate into neural circuits through various mechanisms, including the synaptogenic factor thrombospondin-1 (TSP-1). This new mechanistic understanding of cancer neuroscience allows for novel insights into target discovery. Critically, therapies that modulate neuron-tumor interactions remain agnostic to other oncogenic changes within tumor cells yet may still target fundamental drivers of tumor growth. In line with these findings and controlling for critical confounding variables, we demonstrate a survival benefit associated with gabapentin (an antagonist of TSP-1) following surgical resection of newly diagnosed glioblastoma. This retrospective, multi-institutional cohort study included 1,072 patients, with a discovery cohort of 693 patients and an additional 379 patients from a separate site for external validation. Furthermore, our findings indicate that gabapentin administration is associated with reduced serum TSP-1 levels, suggesting its potential as a future biomarker.

Glioblastoma (GBM) treatment continues to pose a significant challenge in neurosurgery and neuro-oncology, with minimal improvements in life expectancy despite decades of research. This is particularly concerning given the nearly 12,000 new cases diagnosed annually[1]. Of note, GBMs are comprised of a heterogeneous population of cells, which has hampered targeted therapies[2–5]. Recently, the emerging field of cancer neuroscience has uncovered significant insights into how neurons and glial cells can promote and influence tumor development[6,7]. In brief, the electrical, paracrine, and neuronal–glial interactions that drive appropriate neural development,

e.g., myelination, synaptogenesis, and neurotransmitter signaling, can contribute to aberrant neuronal or glial oncogenesis. The underpinnings of these mechanisms are just being discovered, but the core concepts allow for immediate application to current treatment. Of note, GBMs also harbor subpopulations enriched for synaptic formation, which drive tumor progression[8,9]. For instance, oligodendrocyte progenitor cells form unidirectional synapses with neurons, leading to tumor cell invasion and myelination in GBM[6]. Further, functional postsynaptic currents between neurons and GBM tumor cells appear to be mediated through α-amino-3-hydroxy-5-methyl-4-

isoxazolepropionic acid (AMPA) receptors[8–10], which increase calcium influx and malignant cell proliferation. Building on these findings, the discovery of neuron–glioma interactions mediated by AMPA receptors has led to the use of perampanel—an FDA-approved AMPA-blocking anti-epileptic drug—which has been shown to reduce glioma proliferation and invasion[8–10].

In a report from 2023[11], our team demonstrated that the degree of functional connectivity between GBM and the normal brain negatively affected patient survival and cognition. Therein, we employed direct electrophysiologic intraoperative brain recordings in awake patients during a battery of language tasks during tumor resection adjacent to the eloquent speech cortex. When coupled with site-directed biopsies, we demonstrated that such speech tasks activate glioma-infiltrated cortex well beyond those regions that are canonically recruited in healthy brains via remodeled circuitry. Some parts of the tumor were more functionally connected to the brain than others; these areas of high functional connectivity contained cells with heightened responsiveness to neuronal activity. Additionally, this subpopulation of glioma cells secreted thrombospondin-1 (TSP-1), a protein factor that promotes synapse formation and neural circuit remodeling. In fact, prior work utilizing data from the Cancer Genome Atlas (TCGA) research network has suggested a negative association between TSP-1 expression and overall survival in GBM[12].

This mechanistic understanding of neuronal–glial–tumor interactions opens the door to previously unrecognized oncologic therapies for GBM, utilizing commonly prescribed drugs to target synaptic channels and/or other emerging drivers of cancer neuroscience. TSP-1 is a known pro-synaptogenic factor secreted by astrocytes, which binds the α2δ subunit of calcium channels localized at synapses, enabling synapse formation between axons and dendrites[13–16]. In line with these findings, we also demonstrated that gabapentin treatment in preclinical models lead to reduced glioma proliferation when tumor cells were co-cultured with neurons and in mice bearing high-functional connectivity patient-derived xenografts as compared to controls.[11] Gabapentinoids, including gabapentin and pregabalin, are FDA-approved agents commonly prescribed for seizures and neuropathic pain[17]. Their mechanism of action involves binding to the same α2δ subunit of calcium channels that TSP-1 targets[17].

In this work, we investigated if TSP-1 inhibition by gabapentinoids leads to a survival benefit in patients with GBM. We analyzed multi-institutional databases from Mass General Brigham (MGB) and the University of California, San Francisco (UCSF) to determine whether gabapentin administration post-tumor resection is associated with improved overall survival in GBM patients. Additionally, we evaluated the relationship between gabapentin treatment and TSP-1 levels in both serum and tissue samples.

## Results

### Discovery cohort, Mass General Brigham

The MGB discovery cohort consisted of 693 patients with newly diagnosed GBM who underwent surgical treatment at the Brigham and Women's Hospital (BWH)/Dana–Farber Cancer Institute (DFCI) or the Massachusetts General Hospital (MGH) between 2010 and 2022. Following surgery, 103 patients received gabapentin postoperatively at a median dose of 495 (interquartile range [IQR] = 300–900) mg for a median period of 180 (IQR = 59.5–614.0) days, while 590 patients did not receive gabapentin (Supplementary Tables 1–3). The median age at diagnosis was 63 (IQR = 56–72) and 65 (IQR = 57–73) years in the gabapentin and no gabapentin group, respectively. The proportion of females was 53.4% and 44.6% in the gabapentin and no gabapentin groups, respectively. In the gabapentin group, 87.4% of patients were white, 1.9% black, 4.9% Asian/Pacific islander, and 5.8% "other". In the no gabapentin group, 91.4% were white, 2.4% black, 0.9% Asian/Pacific islander, and 5.3% "other". The proportion of patients with O-6-methylguanine-DNA methyltransferase (MGMT) methylation was 45.6% in the gabapentin group and 39.7% in the no gabapentin group. Gross-total resection (GTR) was achieved in 57.3% of patients who received postoperative gabapentin and 44.7% of patients who did not receive postoperative gabapentin. Sub-total resection was achieved in 36.9% of patients in the gabapentin group and in 49.0% of patients in the no gabapentin group, while a tumor biopsy was conducted in 5.8% and 5.9% of the groups, respectively. A combined radiotherapy and chemotherapy regimen was administered to 81.6% of patients in the gabapentin group and 79.1% of patients in the no gabapentin group.

The median overall survival in the entire discovery cohort was 13.0 (95% confidence interval [CI] = 12.1–13.9) months. Median overall survival in patients receiving postoperative gabapentin was 16.0 (95% CI = 13.2–18.8) months compared to 12.0 (95% CI = 11.1–12.9) months for those patients not receiving gabapentin (log-rank test, $p \leq 0.001$). This difference in survival times remained significant in multivariable-adjusted Cox regression analysis (Hazard ratio [HR] = 0.65; 95% CI = 0.51–0.84) (Fig. 1). When adjusting for the use of radiotherapy and

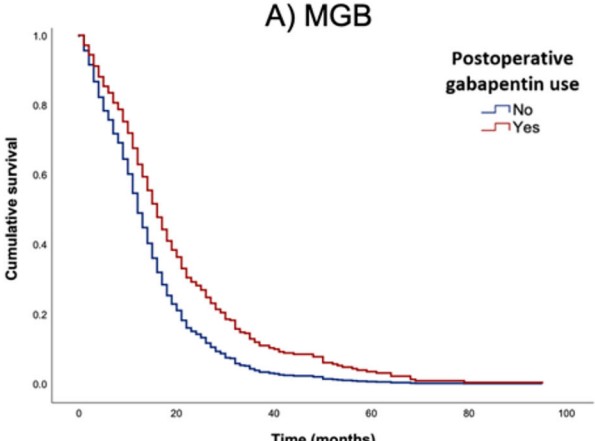
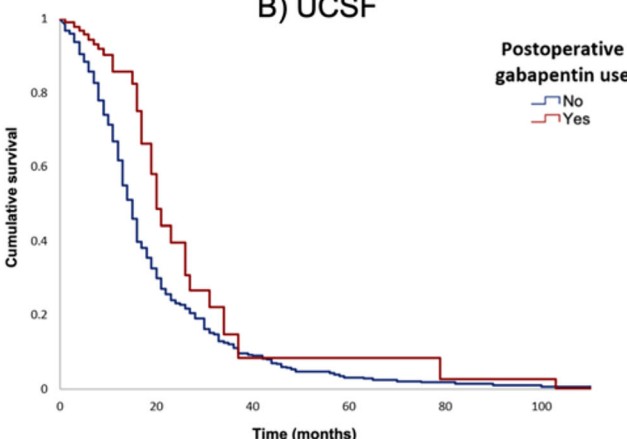

**Fig. 1 | Overall survival of postoperative gabapentin in two independent cohorts.** Cox proportional hazards model controlling for patient age, sex, race, *MGMT* methylation status, postoperative levetiracetam use, other postoperative AED use, EOR, and preoperative KPS was used in both cohorts. **A** Cox plot of overall survival following initial surgical resection comparing patients who received gabapentin to those who did not at MGB. At last follow-up, 74/103 (71.8%) patients

receiving postoperative gabapentin had died, compared to 512/590 (86.8%) patients who did not receive postoperative gabapentin (HR = 0.65; 95% CI = 0.51–0.84). **B** Cox plot of overall survival following initial surgical resection comparing patients in the UCSF cohort who received gabapentin to those who did not (HR = 0.65; 95% CI = 0.44–0.97). Source data is provided as a Source Data file.

chemotherapy, the survival benefit observed after gabapentin use remained (HR = 0.72; 95% CI = 0.56–0.92). Likewise, the survival benefit after gabapentin use remained nearly unchanged when adjusting for left frontal tumor location (i.e., as a proxy for eloquent cortex) (HR = 0.66; 95% CI = 0.51–0.84) (Supplementary Fig. 1). The sensitivity analysis including inverse probability of treatment weighting (IPTW) did not impact the results (weighted HR of gabapentin 0.64; 95% CI = 0.49–0.85).

### Validation cohort, University of California, San Francisco

In the validation cohort from UCSF, there were 379 patients with newly diagnosed GBM. Patient characteristics are presented in Supplementary Table 1. The median age at diagnosis was 58.1 (IQR = 50.1–65.8) years in patients who received gabapentin and 61.4 (IQR = 53.7–68.2) years in patients who did not receive gabapentin. The proportion of females was 50.0% and 37.3% in the gabapentin and no gabapentin cohorts, respectively. In the gabapentin group, 94.4% of patients were white and 5.6% were black. In the no gabapentin group, 91.0% were white, 5.6% black, 1.8% asian/pacific islander, and 1.5% "other". Of the 16 patients in the gabapentin group with known *MGMT* methylation status, 62.5% had methylated tumors. Of 155 patients in the no gabapentin group with known *MGMT* methylation status, 46.5% had methylated tumors. Combined radiotherapy and chemotherapy were given to 80.6% of patients in the gabapentin group and 82.5% of patients in the no gabapentin group. The median total daily dose of gabapentin at UCSF was 600 (IQR = 300–900) mg (Supplementary Table 3). At UCSF, the median overall survival was 20.8 (IQR = 11.7–32.1) months in the gabapentin group and 14.7 (IQR = 8.9–23.5) months in the no gabapentin group (Fig. 1). On multivariable Cox proportional-hazards analysis, treatment with postoperative gabapentin conferred a survival advantage (HR = 0.65, 95% CI = 0.44–0.97).

### Gabapentinoids and tissue/serum TSP-1

Given that gabapentin demonstrated an overall survival benefit, we next sought to understand its relationship to TSP-1 levels in resected tumor and patient serum; this was motivated by our prior work demonstrating that TSP-1 is a regulator of neuronal-driven tumor growth and is inhibited by gabapentin[11]. We performed immunohistochemistry on tumor samples from the initial resection amongst patients treated postoperatively with gabapentin for TSP-1 expression, which was dichotomized into high (≥50% of cells) and low (<50% of cells) levels of expression. Out of the patients that received gabapentin postoperatively, 51 (53.7%; 51/95) patients were high expressors, while 44 (46.3%; 44/95) patients were low expressors. In addition, a control cohort matched by age, sex, and extent of resection was stained; amongst patients that did not receive gabapentin postoperatively, 41 (42.7%; 41/96) patients were high expressors and 55 (57.3%; 55/96) patients were low expressors. The benefit of postoperative gabapentin demonstrated for overall survival did not appear to be associated with the level of TSP-1 expression at initial resection (p = 0.09) (Supplementary Fig. 2).

Serum was available for a subset of patients in the gabapentin validation cohort (n = 9). These nine patients underwent 2:1 propensity score matching based on age, tumor volume, use of chemoradiation, and extent of GBM resection, which was matched to serum samples from no gabapentin patients (n = 18). Critically, the mean serum TSP-1 in gabapentin-treated patients was significantly lower than in untreated patients (10,181 ng/ml vs 17,015 ng/ml, p = 0.017) (Supplementary Fig. 3).

## Discussion

We provide evidence that gabapentin use at the time of initial diagnosis may be associated with increased overall survival in GBM patients, even when controlling for relevant clinical confounders. Although these findings are based on retrospective cohorts, the observed survival benefit–approximately 4 months at MGB and

6 months at UCSF–is comparable to the benefit seen with tumor-treating fields, the most recent adjuvant therapy to receive FDA approval for GBM, which also extends survival by roughly 6 months[18].

It is prudent to note that not all findings reported in the literature are congruent with our results. A recent paper by Ryu et al. found that anticonvulsants conferred a survival benefit in their retrospective cohort, but gabapentinoids were not among those that were significant[19]. This study was smaller and found that pregabalin, which also binds to α2δ like gabapentin, trended towards significance. Furthermore, the predominantly Asian study population may not be fully representative of the primarily European ancestry population in the present paper. The known differences in glioma survival across different racial and ethnic groups could account for this discrepancy[20].

Although our study is retrospective, we accounted for key survival-related confounders and minimized the risk of selection bias by replicating our findings in two independent cohorts. Additionally, the overall number of patients is large, with 139 patients ultimately having received gabapentin post-surgery across the two cohorts. Further, these clinical data are based on the molecular mechanism put forth by our group[11]. However, larger, randomized, multicenter studies will ultimately be needed to confirm the findings presented and, in doing so, provide more robust evidence for the use of neuromodulators/anticonvulsant medications as adjuvant therapies in GBM, and to identify optimally responsive sub-populations of patients (e.g., based on TSP-1 quantification at initial resection, serum TSP-1, or structural/functional connectivity). Additionally, the absence of stratified molecular subanalyses and the inability to validate tissue-level target engagement further underscores the need for deeper mechanistic investigations. These limitations also extend to the precise monitoring of dosing, which in the current study was constrained by its retrospective design and reliance on medical records. Aligned with this approach, ongoing clinical trials are prospectively investigating the effects of brain-penetrating anti-synaptogenic and glutamatergic drugs in addition to the Stupp protocol, with enrollment currently underway (NCT05664464)[21]. Additionally, our group is in the final stages of planning a prospective randomized study to assess the impact of gabapentin on overall survival in GBM with standardized dosing regimens to evaluate dose-response relationships, detailed molecular profiling (e.g., transcriptomics and direct measures of target engagement), and longitudinal follow-up to assess progression-free survival. The value of this study lies in its provision of a clinical and scientific rationale for undertaking resource- and labor-intensive prospective, randomized clinical trials, while also highlighting the potential of targeting cancer neuroscience with therapeutic interventions[7,22].

Crucially, administration of gabapentin, a known to inhibitor[11] of TSP-1, was associated with reduced circulating serum TSP-1 levels as compared to matched controls. This preliminary finding provides further evidence that gabapentin-mediated TSP-1 inhibition (corresponding to reduction of peripheral TSP-1) impairs the proliferation of GBM, which in previous work targeted cell subpopulations within intratumoral regions with high connectivity. Consequently, TSP-1 may serve as a prognostic indicator for drug response and GBM's ability to remodel neural circuits; ultimately, this pattern may be observed in brain tissue via focal/serial biopsies and in re-resected tissue when compared to primary tumors[23]. While these pilot TSP-1 findings are encouraging, it is important to note that individual patient serum TSP-1 changes following gabapentin administration were not evaluated. Furthermore, TSP-1 is expressed by various peripheral sources in addition to the CNS[24]. As such, a causal relationship between gabapentin administration and peripheral TSP-1 expression could not be established based on these retrospective data. We also did not establish a clear relationship between the TSP-1 expression level of the tumor and gabapentin response or prognosis in this retrospective study. Nevertheless, the observed trend is promising, suggesting serum TSP-1 levels may serve as a potential biomarker for therapeutic

response. The extent to which TSP-1 expression and other factors modulate response to gabapentin will be evaluated in our future prospective work.

The finding that increased functional connectivity is associated with worse survival underscores the complex challenge clinicians and surgeons face in glioma treatment. It requires a delicate balance between achieving maximal tumor resection and preserving "eloquent" brain regions to minimize neurologic deficits, which can severely affect quality of life [25]. However, emerging evidence suggests that areas with heightened functional connectivity may actually promote tumor growth, indicating that regions with the most significant functional connectivity could also be the sites of more aggressive tumor proliferation. This insight may open the door to novel treatment approaches that target activity-dependent mechanisms of GBM proliferation, offering the potential to enhance both oncologic outcomes and functional preservation in glioma patients [7,22]. Taken together with prior evidence, this work provides further support for the repurposing of gabapentinoids and for TSP-1 inhibition for the treatment of GBM, which will be further evaluated in prospective trials.

## Methods

### Study design and patient population

The present study was completed in accordance with the Strengthening the Reporting of Observational Studies in Epidemiology (STROBE) guidelines. To assess the impact of gabapentin on overall survival, a multi-institutional retrospective cohort study was conducted at three major academic institutions. The discovery cohort, from the MGB system in Boston, MA (USA), included patients undergoing surgery between 2010 and 2022 at BWH/DFCI or MGH. The validation cohort included patients treated at UCSF in San Francisco, CA (USA), between 1997 and 2017. The study protocol was approved by the Institutional Review Board (IRB) of BWH (IRB no. 2015P002352) and UCSF (IRB no. 17-23215).

In both cohorts, GBM patients (wild-type isocitrate dehydrogenase [*IDH*-wt]; 2021 World Health Organization Classification[26]) aged >18 years were included. Patients were excluded if the primary surgery was performed outside of the included hospitals and if recurrent GBM was present.

### Data collection and definitions

Data were acquired retrospectively through a review of electronic medical records. At all institutions, demographic, clinical, and surgical data were collected, including age, sex, and race/ethnicity, tumor lobe/hemisphere, Karnofsky performance status (KPS) score, *MGMT* gene promoter methylation status, extent of resection (EOR), and compliance with adjuvant temozolomide and radiation therapy. Data was collected using Microsoft Excel Version 2208. In both cohorts, the dose and duration of gabapentin prescription were recorded. Additionally, at both institutions, the presence of seizures and the use of antiepileptic drugs were collected. The validation cohort at UCSF collected tumor volumes for each patient.

In the discovery cohort, EOR was determined according to radiologist reports. In the validation cohort, EOR and tumor volumes were determined in the following manner. Pre- and post-operative tumor volumes were quantified using BrainLab Smartbrush software (Brainlab, Munich, Germany). Pre-operative MRI scans were obtained within 24 h before resection, and post-operative scans were all obtained within 72 h post-resection. Total contrast-enhancing (CE) and non-enhancing (NE) tumor volumes were measured at pre-operative and post-operative times. The total CE tumor volume was measured on T1-weighted post-contrast images, and the non-enhancing tumor volume was measured on T2 or fluid-attenuated inversion-recovery (FLAIR) sequences. Manual segmentation was performed with region-of-interest analysis, "painting" inclusion regions based on FLAIR sequences from pre- and post-operative MRI scans to quantify tumor volume. The EOR was calculated as:

$$\frac{(\text{pre}-\text{operative tumor volume}) - (\text{post}-\text{operative tumor volume})}{(\text{pre}-\text{operative tumor volume})} \times 100\%$$

(1)

Multifocal or multicentric disease was defined as noncontiguous areas of disease based on T1-weighted post-contrast images or FLAIR sequences. All multifocal lesions were measured separately and combined for a single volumetric measurement. Volumetric measurements were made blinded to clinical outcomes. All evaluable patients in the cohort had available preoperative and postoperative MRI scans for analysis. FLAIR pre- and post-operative MRIs were carefully compared alongside DWI sequences before including each region in the volume segmentation to ensure that the post-operative FLAIR signal was not surgically-induced edema or ischemia.

Overall survival was the primary outcome of this study and was defined as the time from the date of surgical resection of GBM until the date of death. Censoring was performed on the day of the last recorded visit before the end of the study (May 5, 2023, at the MGB discovery cohort and December 9, 2018, at the UCSF validation cohort) or at the end of the study period.

### Serum analysis

Gabapentin-treated patients at the UCSF validation cohort with available serum had a propensity score that was matched (2:1) based on the key prognostic factors of age at diagnosis, tumor volume, and EOR, matched to untreated patients. Serum TSP-1 levels were measured by enzyme-linked immunosorbent assay (ELISA). Peripheral blood samples from newly diagnosed patients with GBM were collected and allowed to clot for 30 min at room temperature before centrifugation for 15 min at 1000*g*. The serum was stored at −80 °C until analysis. The TSP-1 level was determined using the Quantikine immunosorbent assay kits according to the manufacturer's instructions (R&D Systems). Unpaired two-tailed Student's *t*-test was used to compare mean serum TSP-1 levels between the gabapentin-treated and untreated cohorts.

### Immunohistochemistry

Resected tumor samples from patients treated with gabapentin at the MGB discovery cohort were stained for TSP-1 expression levels and scored from 0 to 4 (0 ≤ 5% stained, 1 = 5−25% stained, 2 = 25−50% stained, 3 = 50−75% stained and 4 ≥ 75% stained) (Supplementary Fig. 4); which was dichotomized into high (≥50% of cells) and low (<50% of cells). A 1:1 propensity score matched control group of patients not on gabapentin was generated and stained. These patients were matched on EOR, age, and sex. These were stained and scored in the same fashion.

### Statistical models

A log-rank test was used to assess the statistical significance in months of overall survival of patients who were treated with postoperative gabapentin versus those who were not. Multivariable Cox regression analyses were used to assess the effect of potential confounders on overall survival; all Cox regression analyses were adjusted a priori for possible confounders for survival in patients with GBM. In both cohorts these included age (continuous), sex (male, female), race (white, other), *MGMT* methylation status (methylated, unmethylated, partially methylated being considered unmethylated), EOR (gross-total, sub-total, biopsy), postoperative levetiracetam use (yes, no), postoperative other AED use (yes, no) and preoperative KPS (≥70, <70). In an additional multivariable model, we adjusted for additional potential confounders in the Cox regression model, such as radiotherapy and chemotherapy, as well as left frontal tumor location. We also performed a sensitivity analysis using IPTW. The propensity scores were calculated by performing a multivariable logistic regression using the variables that we adjust for in the Cox regression model as independent variables (age,

sex, race, *MGMT* methylation status, EOR, postoperative levetiracetam use, postoperative other AED use and preoperative KPS) and postoperative gabapentin use as the dependent variable. The propensity scores were inverted and included as weights in the multivariable Cox regression model. Results of the sensitivity analysis are presented in the Supplementary Table 4. To assess whether TSP-1 tumor expression modified the association between gabapentin and overall survival, an interaction term between gabapentin and TSP-1 was included.

All analyses were performed using IBM Statistical Package for the Social Sciences (SPSS) version 29.0.1.0 and R Statistical Software version 4.4.0.

### Reporting summary
Further information on research design is available in the Nature Portfolio Reporting Summary linked to this article.

### Data availability
The analyzed data generated in this study are available under restricted access for research purposes. The raw patient data are protected and are not available due to data privacy laws. Access can be obtained by submitting requests to the corresponding authors (jbernstock@bwh.harvard.edu and shawn.hervey-jumper@ucsf.edu). The Institute Review Boards at Brigham and Women's Hospital and UCSF will then proceed to review specific requests. The processed data generated in this study are provided in the Supplementary Information/Source Data file. Source data are provided with this paper.

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

### Acknowledgements

L.S. is funded by the Deutsche Forschungsgemeinschaft (DFG, German Research Foundation)—534053725.

### Author contributions

Conceptualization, J.D.B.; methodology, J.D.B., M.M., J.V.E.G., D.M.M., P.A.V., P.H., V.S.A., S.K., J.A.C., and H.A.; writing—original draft preparation, J.D.B., M.M., and J.V.E.G.; writing—review and editing, J.D.B., M.M., J.V.E.G., D.M.M., P.A.V., P.H., V.S.A., S.K., J.A.C., H.A., B.R.J., M.N., K.G., C.N.G., A.D., A.B.L.V., A.K., L.S., J.K., S.O., Y.E.S., R.A.M., A.D., J.F.G., S.I., F.A.G., S.S., O.A., Y.L., B.D.C., M.D.H., M.P.M., Y.O.m G.K.F., E.A.C., P.P.P., T.R.S., and S.L.H.J.; visualization, J.V.E.G.; supervision, J.D.B., E.A.C., P.P.P., T.R.S., and S.L.H.J. All authors have read and agreed to the published version of the paper.

### Competing interests
The authors have no personal, financial, or institutional interest in any of the drugs, materials, or devices described in this article. Joshua D. Bernstock has an equity position in Treovir Inc. and UpFront Diagnostics. Dr. Bernstock is also a cofounder of Centile Bioscience and is on the NeuroX1 and QV Bioelectronics scientific advisory boards. Jason A. Chen is a cofounder and holds equity in Verge Genomics. Dr. Chen is also an advisor and holds equity in Gravity Medical Technology. Florian A. Gessler received honoraria from Signus, BBraun, Aesculap, and Astra-Zeneca. Gregory K. Friedman is supported by Eli Lilly and Company and Pfizer through contracts to UAB. Dr. Smith reports being a cofounder and equity holder of PheBe Health Inc.

## Additional information

Joshua D. Bernstock or Shawn L. Hervey-Jumper.

**Peer review information** *Nature Communications* thanks Jian-Guo Zhou
and the other, anonymous, reviewer(s) for their contribution to the peer
review of this work. A peer review file is available.

**Publisher's note** Springer Nature remains neutral with regard to
jurisdictional claims in published maps and institutional affiliations.

[1]Department of Neurosurgery, Brigham and Women's Hospital, Harvard Medical School, Boston, MA, USA. [2]Department of Neurosurgery, Boston Children's Hospital, Harvard Medical School, Boston, MA, USA. [3]David H. Koch Institute for Integrative Cancer Research, Massachusetts Institute of Technology, Cambridge, MA, USA. [4]Department of Neurological Surgery, University of California San Francisco, San Francisco, CA, USA. [5]Computational Neuroscience Outcomes Center (CNOC), Department of Neurosurgery, Brigham and Women's Hospital, Harvard Medical School, Boston, MA, USA. [6]Department of Pathology, Brigham and Women's Hospital, Harvard Medical School, Boston, MA, USA. [7]Department of Neurosurgery, University of Texas Medical Branch, Galveston, TX, USA. [8]Department of Neurosurgery, Rostock University Medical Center, Rostock, Germany. [9]School of Pharmacy, Massachusetts College of Pharmacy and Health Sciences (MCPHS) University, Boston, MA, USA. [10]Department of Neurology, University of California San Francisco, San Francisco, CA, USA. [11]Department of Neurology, Brigham and Women's Hospital, Harvard Medical School, Boston, MA, USA. [12]Department of Neurosurgery, Massachusetts General Hospital, Harvard Medical School, Boston, MA, USA. [13]Department of Radiation Oncology, Miami Cancer Institute, Baptist Health South Florida, Miami, FL, USA. [14]Herbert Wertheim College of Medicine, Florida International University, Miami, FL, USA. [15]Division of Pediatrics, Neuro-Oncology Section, The University of Texas MD Anderson Cancer Center, Houston, TX, USA. [16]Weill Institute for Neurosciences, University of California San Francisco, San Francisco, CA, USA. [17]These authors contributed equally: Joshua D. Bernstock, Mulki Mehari, Jakob V. E. Gerstl. [18]These authors jointly supervised this work: E. A. Chiocca, Pier Paolo Peruzzi, Timothy R. Smith, Shawn L. Hervey-Jumper. ✉e-mail: jbernstock@bwh.harvard.edu; shawn.hervey-jumper@ucsf.edu

