## [Transparent Peer Review file · Nature Communications]

Gabapentinoids confer survival benefit in human glioblastoma

Corresponding Author: Dr Joshua Bernstock

Version 1:

Reviewer comments:

Reviewer #3

(Remarks to the Author)

This study by Bernstock et al. investigates an important clinical question regarding the potential therapeutic benefit of gabapentin in glioblastoma patients. The authors present intriguing retrospective data suggesting improved survival in gabapentin-treated patients across two independent cohorts, which represents a potentially significant finding given the limited treatment options for this devastating disease.

Strengths:

- Large patient cohorts (693 discovery + 379 validation patients) from major academic centers
- Consistent survival benefit observed across independent cohorts
- Preliminary evidence of target engagement through serum TSP-1 measurements
- Clear clinical relevance given gabapentin's established safety profile and immediate potential for repurposing
- Robust statistical methodology including controls for confounding variables

Limitations:

Lack of comprehensive molecular characterization:

- No methylation array analyses
- No single-cell or spatial transcriptomics
- Limited ability to identify molecular predictors of response
- Insufficient understanding of potential selection biases

Inadequate demonstration of target engagement:

- Only serum TSP-1 data provided without tissue-level validation
- No analysis of differential effects on tumor vs microenvironment cells
- Limited understanding of synaptic density changes and pathway activation
- Small number of matched primary/recurrent specimens

Variable treatment parameters:

- Heterogeneous dosing and duration of gabapentin therapy
- Limited ability to establish dose-response relationships
- Unclear optimal treatment regimen

Mechanistic gaps:

- Insufficient correlation between functional connectivity and treatment response
- Limited understanding of which patient subgroups may benefit most
- Unclear relationship between TSP-1 expression and clinical benefit

Recommendations:

While the clinical observations are compelling and potentially impactful, several key analyses would strengthen the manuscript:

- Even with limited tissue availability, targeted molecular analyses of available samples could provide crucial insights into response mechanisms
- More detailed analysis correlating tumor location, functional connectivity, and treatment response
- Better characterization of the relationship between TSP-1 expression and clinical benefit
- Clearer articulation of the limitations and need for prospective validation

Conclusion:

The manuscript presents potentially important clinical findings that warrant serious consideration. However, the lack of molecular and mechanistic characterization limits our ability to fully interpret the results and optimize future clinical applications. While acknowledging the challenges of retrospective analyses, additional molecular studies would significantly strengthen the manuscript's impact and clinical utility.

The central question is whether the clinical significance of the findings outweighs the current mechanistic limitations. Given the urgent need for new therapeutic approaches in glioblastoma, there is merit to publishing these findings to enable further investigation. However, this should be balanced against the risk of prematurely promoting a therapeutic approach without fully understanding its biological basis.

A potential compromise would be to:

- More explicitly acknowledge the current limitations
- Better define the key questions for prospective investigation
- Provide preliminary molecular analyses where possible with available samples
- Frame the findings as hypothesis-generating rather than definitive

This would allow the field to build on these observations while maintaining appropriate scientific rigor.

Reviewer #4

(Remarks to the Author)

This is the third version of the manuscript. I commend the authors for their dedication and hard work in refining the study. Below, I outline my key comments:

A central issue arises from the finding that gabapentin acts as an antagonist of TSP-1, yet the authors report that overall survival (OS) is not significantly associated with TSP-1 expression levels at initial resection ($p = 0.09$; Supplementary Figure 2). This result appears to contradict the authors' hypothesis. I recommend that the authors conduct a more detailed analysis by comparing OS and progression-free survival (PFS) across three distinct groups: (1) TSP-1 positive patients treated with gabapentin, (2) TSP-1 positive patients not treated with gabapentin, and (3) TSP-1 negative patients. This subgroup analysis may provide greater insight into the role of TSP-1 expression in mediating gabapentin's effects on survival outcomes. Despite reviewing the authors' response, I remain unclear on this issue.

Regarding the second point, the authors acknowledge the concern in principle and have outlined plans to design future prospective studies to address these goals. However, further clarification on how these studies will be structured and how they will address the current limitations would be helpful.

We thank the reviewers for taking the time to review the revised manuscript and provide thoughtful comments. Reviewers' comments are presented in **bold**, our responses are in **green**, text that was in the original submission is in **blue**, and new text is in **red**.

REVIEWER 1

Comment 1.1:

Reviewer #1 (Remarks to the Author):

1. Can you clarify whether or not the serum TSP-1's only source is central (related to tumor or the CNS) and that no TSP-1 is known to have a peripheral source. Is there a correlation between serum TSP-1 and tumor expression of TSP-1. Is the impact of gabapentinoids only related to peripheral changes in serum TSP-1

Authors' Response:

We thank the Reviewer for highlighting this point. TSP-1 has multiple peripheral sources in addition to central sources,¹ which may confound the observed association between TSP-1 and gabapentin. This has been acknowledged and incorporated into the study's limitations. Since tumor-derived TSP-1 expression was only available from MGB, and peripheral TSP-1 data was exclusive to UCSF, direct correlation between these measures was not feasible.

Changes to the text:

“While these pilot TSP-1 findings are encouraging, it is important to note that TSP-1 is expressed by various peripheral sources in addition to the CNS.¹ As such, a causal relationship between gabapentin administration and peripheral TSP-1 expression could not be established based on these retrospective data. Nevertheless, the observed trend is promising, suggesting TSP-1 may serve as a potential biomarker for therapeutic response.” (Lines 242-248).

Comment 1.2:

2. If I understand correctly, the effects size suggested by this manuscript show HR of .6 or so, this lessens when considering those who receive RT/Chemo to HR 0.72. Why not just provide those where patients have had RT/Chemo as this would level set the patient population comparisons. This will also help with understanding the prognostic factors, and if they are truly balanced, as they are only known when one considers RT/Chemo. Additionally please provide the multivariable evaluation on the RT/Chemo only populations.

Authors' Response:

The Reviewer raises an important and insightful point. In the IPTW sensitivity analysis, RT/Chemo was associated with a marginal increase in the HR, potentially suggesting that receiving RT/Chemo might predict a poorer response to gabapentin. However, an opposite trend emerged when we performed the analysis as suggested in the initial submission—namely, conducting separate Cox regressions for RT and Chemotherapy:

We chose to include stratified analyses rather than separate Cox regressions, as presented here, due to the likely low power of the “No chemotherapy” arm (given that this is not the clinical standard/small number of patients). In light of the absence of clear trends, we prefer to adopt a conservative approach regarding conclusions about potential predictors. Therefore, we have not modified the analysis or the conclusions in the manuscript. However, to address this point, we have added the below statement to the discussion.

Changes to the text:

“The extent to which TSP-1 expression, and other factors, modulate response to gabapentin will be evaluated in our future prospective work.” (Lines 248-250).

Comment 1.3

3. Can you clarify if the measurement of serum TSP-1 is impacted by time. was there any significant difference as to when these samples were obtained for each group and when run. Some of these sample were very old and other relatively more contemporary.

Authors’ Response:

Thank you for bringing this to our attention. Upon assessing the temporal trends, we found no evidence of a correlation between time and TSP-1 concentration.

Comment 1.4

4. Electronic medical records have evolved dramatically from 2010 to 2022 and in particular from 1997 to 2017, including whether the data on medication start and stop date are structured data or text and electronic prescriptions are only a relatively recent phenomena. Since understanding when gabapentin started, stopped and the dose used seems critical, this reviewer, who has over 30 years of experience with investigations from medical records remains unconvinced that these data are accurate and reliable.

Authors' Response:

We acknowledge that determining exact dates, particularly for earlier time points, can be challenging. We have explicitly added this limitation to the manuscript. This issue may, for instance, help explain the absence of a dose-response relationship observed in our analysis. However, errors regarding whether gabapentin was prescribed at all—the core aspect of our primary analysis—are less likely, given the comprehensive oncologic workup conducted in both cohorts. Moving forward, prospective studies, should aim to more closely evaluate dose-response relationships to address this limitation; such work is being planned by our own group and is due to commence in 2025.

Changes to the text:

“This limitation also extends to the precise monitoring of dosing, which in the current study was constrained by its retrospective design and reliance on medical records.” (Lines 227-228).

Comment 1.5

5. Were any patients on gabapentinoids when undergoing the initial surgical resection. If so, did you evaluate and compare sample with and without exposure?

Authors' Response:

Unfortunately, only three patients in the discovery cohort were on preoperative gabapentin but not postoperative gabapentin. As a result, conducting such an analysis was not statistically possible.

Comment 1.6

6. For supplementary table 1 can you create:

- 1. Another column on this table indicating how many days from initiating RT/chemo, did gabapentin start and indicate with negative days whether any of these patents had received gabapentin before/or between surgical resection and RT?**
- 2. Indicate on another column whether or not RT chemo was used**
- 3. Indicate on another column the survival in days and if these survival data are censored.**

Authors' Response:

We have updated the table as per the Reviewer's suggestion. However, "days from initiating chemotherapy to starting gabapentin" were not included due to the limitations highlighted in **Comment 1.4**.

Changes to the text:

Supplementary Table 1: Gabapentin treatment dosing, timing, concurrent chemotherapy/radiotherapy, and survival in GBM patients within the MGB discovery cohort.

Patient Sequence	Duration of Gabapentin (in days)	Weighted Average Daily Dosage (in mg)	Preoperative Gabapentin	Postoperative Radiotherapy	Postoperative Chemotherapy	Postoperative Survival (in months)
Patient 1	148	300	No	Yes	Yes	95
Patient 2	111	1354	No	Yes	Yes	14
Patient 3	335	920	No	Yes	Yes	11
Patient 4	Unknown	600	Yes	Yes	Yes	18
Patient 5	82	900	Unknown	Yes	Yes	5
Patient 6	Unknown	600	Yes	Yes	Yes	4
Patient 7	Unknown	300	Yes	Yes	Yes	32
Patient 8	40	300	No	Yes	Yes	28
Patient 9	808	1526	No	Yes	Yes	50
Patient 10	11	200	Yes	Yes	Yes	5
Patient 11	26	1800	Unknown	Yes	Yes	25
Patient 12	1605	300	Unknown	Yes	Yes	16
Patient 13	539	300	Yes	Yes	Yes	32
Patient 14	141	600	Unknown	Yes	Yes	12
Patient 15	Unknown	Unknown	Unknown	Yes	Yes	14
Patient 16	921	576	Unknown	Yes	Yes	34
Patient 17	53	1200	Unknown	Yes	Yes	39
Patient 18	240	900	Unknown	Yes	Yes	9
Patient 19	1090	882.6	Yes	Yes	Yes	37
Patient 20	67	636	No	Yes	Yes	18
Patient 21	664	1200	Yes	Yes	Yes	4
Patient 22	86	1794	No	Yes	Yes	13
Patient 23	258	900	Yes	No	Yes	4
Patient 24	40	818	Yes	No	No	2
Patient 25	159	300	No	Yes	Yes	10
Patient 26	Unknown	300	No	Yes	Yes	Alive
Patient 27	202	1800	No	Yes	Yes	13
Patient 28	1	200	No	Yes	Yes	Alive
Patient 29	1	300	No	No	No	3
Patient 30	38	1642	No	Yes	Yes	42
Patient 31	74	900	Yes	Yes	Yes	9
Patient 32	348	190	No	Yes	Yes	21
Patient 33	46	600	No	Yes	Yes	19
Patient 34	413	235	Yes	Yes	Yes	4
Patient 35	2	600	No	Yes	Yes	30
Patient 36	70	100	No	Yes	Yes	30
Patient 37	1604	2297	Yes	Yes	Yes	2
Patient 38	Unknown	900	Yes	Yes	Yes	26
Patient 39	796	434	Yes	Yes	Yes	27
Patient 40	176	900	No	Yes	Yes	Alive
Patient 41	21	100	No	Yes	Yes	Alive
Patient 42	1215	600	Yes	No	No	Alive
Patient 43	Unknown	300	No	Yes	Yes	14
Patient 44	9	900	No	Yes	Yes	18

Patient 45	1111	300	Yes	Yes	Yes	15
Patient 46	255	1068	No	Yes	Yes	12
Patient 47	746	272	Yes	Yes	Yes	Alive
Patient 48	1	300	No	Yes	Yes	5
Patient 49	189	100	Yes	Yes	Yes	Alive
Patient 50	280	300	No	Yes	Yes	10
Patient 51	217	400	No	Yes	Yes	21
Patient 52	475	1144	No	Yes	No	22
Patient 53	2342	2100	Yes	No	No	3
Patient 54	96	800	No	Yes	No	7
Patient 55	604	300	No	Yes	Yes	26
Patient 56	Unknown	300	No	Yes	Yes	16
Patient 57	190	200	No	Yes	Yes	15
Patient 58	43	900	Yes	No	No	1
Patient 59	83	506	Yes	Yes	Yes	10
Patient 60	Unknown	600	Yes	Yes	No	16
Patient 61	1662	600	Yes	No	No	Alive
Patient 62	30	900	No	Yes	Yes	18
Patient 63	63	300	Yes	Yes	Yes	11
Patient 64	470	300	Yes	Yes	Yes	27
Patient 65	2127	300	Yes	Yes	Yes	18
Patient 66	86	1664	No	Yes	Yes	16
Patient 67	9	100	No	Yes	Yes	Alive
Patient 68	297	787	No	Yes	No	Alive
Patient 69	1848	100	Yes	Yes	No	12
Patient 70	1458	300	Yes	Yes	Yes	Alive
Patient 71	Unknown	600	No	Yes	Yes	Alive
Patient 72	Unknown	100	No	Yes	Yes	10
Patient 73	180	1200	No	Yes	Yes	Alive
Patient 74	Unknown	300	Yes	Yes	Yes	19
Patient 75	189	840	No	Yes	No	21
Patient 76	143	900	Yes	Yes	Yes	Alive
Patient 77	675	300	Yes	Yes	Yes	Alive
Patient 78	877	640	Yes	Yes	Yes	10
Patient 79	624	200	Yes	Yes	Yes	18
Patient 80	Unknown	300	Yes	Yes	Yes	Alive
Patient 81	Unknown	200	Yes	Yes	Yes	11
Patient 82	879	900	Yes	Yes	Yes	Alive
Patient 83	106	300	No	Yes	No	12
Patient 84	19	300	No	Yes	Yes	2
Patient 85	58	484	No	Yes	Yes	8
Patient 86	Unknown	Unknown	Yes	Yes	Yes	13
Patient 87	69	774	No	Yes	Yes	3
Patient 88	1802	600	Yes	Yes	Yes	Alive
Patient 89	500	379	Yes	Yes	Yes	13
Patient 90	Unknown	100	No	Yes	Yes	Alive

Patient 91	2494	300	No	Yes	No	5
Patient 92	99	900	No	Yes	Yes	Alive
Patient 93	151	300	No	Yes	Yes	Alive
Patient 94	61	300	Yes	Yes	Yes	4
Patient 95	501	900	Yes	Yes	Yes	Alive
Patient 96	382	300	No	Yes	Yes	Alive
Patient 97	30	300	No	Yes	Yes	Alive
Patient 98	Unknown	Unknown	No	Yes	No	Alive
Patient 99	2015	400	Yes	No	No	3
Patient 100	165	300	Yes	Yes	No	Alive
Patient 101	53	300	No	Yes	Yes	Alive
Patient 102	50	700	No	No	Yes	Alive
Patient 103	1	600	No	Yes	Yes	Alive

Comment 1.7

7. For supplementary table 2 can you provide the same set of columns as are currently in supplementary table 1 in addition to the information asked for in #6 above.

Authors' Response:

We have updated the table as per the Reviewer's suggestion. As per the above, "days from initiating chemotherapy to starting gabapentin" were not included.

Changes to the text:

Supplementary Table 2: Gabapentin treatment dosing, timing, concurrent chemotherapy/radiotherapy, and survival in GBM patients within the UCSF validation cohort.

Patient (N=36)	Dose (mg)	Daily frequency	Total daily dose (mg)	Treatment with chemoradiation	Overall survival (days)	Censored (Yes, No)
Patient 1	100	1	100	Both	1038	No
Patient 2	100	1	100	Chemotherapy only	364	No
Patient 3	100	1	100	Neither	2438	No
Patient 4	100	2	200	Both	267	No
Patient 5	100	3	300	Both	4862	Yes
Patient 6	300	1	300	Neither	1459	Yes
Patient 7	100	3	300	Both	635	No
Patient 8	300	1	300	Both	508	No
Patient 9	300	1	300	Both	598	No
Patient 10	300	1	300	Both	843	No
Patient 11	300	1	300	Both	531	No
Patient 12	300	1	300	Both	649	No
Patient 13	100	3	300	Both	1144	No
Patient 14	100	3	300	Both	977	Yes
Patient 15	100 AM, 300 PM	1, 1	400	Both	482	No
Patient 16	600	1	600	Both	4410	Yes
Patient 17	300	2	600	Radiation only	107	No

Patient 18	300	2	600	Both	969	Yes
Patient 19	300	2	600	Chemotherapy only	290	No
Patient 20	300	3	900	Both	605	No
Patient 21	300	3	900	Both	3158	No
Patient 22	300	3	900	Both	961	No
Patient 23	300	3	900	Both	708	No
Patient 24	300	3	900	Both	629	No
Patient 25	300	3	900	Neither	144	No
Patient 26	300	3	900	Both	56	No
Patient 27	300	3	900	Both	187	No
Patient 28	300	3	900	Both	673	Yes
Patient 29	300	3	900	Both	812	No
Patient 30	900	2	1800	Both	523	No
Patient 31	600	3	1800	Radiation only	353	No
Patient 32	1200	2	2400	Both	167	No
Patient 33	600	4	2400	Both	811	No
Patient 34	Unknown	Unknown	Unknown	Both	488	No
Patient 35	Unknown	Unknown	Unknown	Both	234	No
Patient 36	Unknown	Unknown	Unknown	Both	2607	No

REVIEWER 2

NA

REVIEWER 3

Comment 3.1:

Reviewer #3 (Remarks to the Author):

The study by Bernstock et al. investigates the potential survival benefit of gabapentin in glioblastoma patients through a retrospective analysis. The authors have revised their manuscript by excluding IDH-mutant tumors, adding a validation cohort from UCSF, and providing data on serum TSP-1 levels. While these modifications strengthen certain aspects, concerns about the molecular characterization of their cohorts and demonstration of target engagement remain.

Major Points:

1. The authors have not provided the requested comprehensive molecular characterization of their patient cohorts. The implementation of methylation array analyses, single-cell or spatial transcriptomics from their available tissue blocks would be essential to identify what potentially differentiates the gabapentin patient cohort from its control group. This analysis is particularly important given the retrospective nature of the study and could provide crucial insights into potential response predictors and/or differences in the patient cohorts that received gabapentin.

Authors' Response:

Thank you for this valuable feedback. We fully agree that a comprehensive molecular characterization of the patient cohorts, including methylation array analyses, single-cell, or spatial transcriptomics, would provide critical insights into potential predictors of response and differences between cohorts. Unfortunately, the retrospective nature of our study and resource limitations, including the availability of preserved tissue blocks suitable for these advanced analyses, prevented us from conducting such

evaluations at this time.

That said, we acknowledge the importance of these techniques and strongly agree that future studies should prioritize integrating molecular analyses to better understand the biological mechanisms underlying response differences in gabapentin-treated patients. We have included this as a limitation in the discussion and emphasized the need for prospective studies incorporating these approaches to enhance the interpretation and applicability of findings. Again, and as highlighted in the response to **Comment 1.4**, such work is being planned by our own group which we have highlighted in the changes to the manuscript below.

Additionally, we would like to underscore the clinical significance of our current findings (confirmed in multiple centers of excellence). Gabapentin emerges as a potentially actionable drug ready for repurposing in an intractable disease, demonstrating significant potential to improve survival with minimal side effects. While molecular characterization remains important, the immediate clinical relevance of these results should take precedence, offering a promising avenue for improving patient outcomes.

Changes to the text:

“The extent to which TSP-1 expression, and other factors, modulate response to gabapentin will be evaluated in our future prospective work.” (Lines 248-250).

Comment 3.2:

2. While the authors show decreased serum TSP-1 in gabapentin-treated patients, this provides only indirect evidence of drug effect. How is this correlated to actual levels of TSP-1 in tissue and in which cells? How are tumor cells differentially affected as compared to the cells of the microenvironment? The crucial question of target engagement within the tumor tissue remains inadequately addressed. A thorough analysis of matched primary and recurrent specimens, examining TSP-1 expression patterns, synaptic density, and pathway activation in regions of high functional connectivity would significantly strengthen their mechanistic claims.

Authors' Response:

We agree in principle and have designed future prospective studies to address these goals. However, in the interim, we believe it is crucial to share this actionable clinical data to assist patients in desperate need. For further details, please refer to our response to **Comment 3.1**.

Comment 3.3:

3. The molecular stratification of patients who received gabapentin versus those who did not remains superficial. The authors have provided basic clinical characteristics but have not leveraged their tissue resources to perform detailed molecular analyses that could explain the observed survival difference. This is particularly important given the variable dosing and duration of gabapentin therapy in their cohort.

The authors acknowledge having fewer than 20 re-resected specimens, which severely limits their ability to demonstrate target engagement over time. This small sample size prevents meaningful assessment of molecular changes following treatment and leaves crucial questions about the mechanism of action unanswered.

Authors' Response:

See response to **Comments 3.1 and 3.2**.

Comment 3.4:

4. Target engagement will be one important aspect of the overall analyses. Which cells have been really targeted and how are they affected? How can the authors make this point from the retrospective analyses?

Authors' Response:

See response to Comments 3.1 and 3.2.

Comment 3.5:**Minor Points:**

1. A more detailed analysis correlating tumor location with functional connectivity and treatment response would be valuable.

Authors' Response:

See response to Comments 3.1 and 3.2.

Comment 3.6:

2. The rationale for gabapentin administration in different patients should be more thoroughly explored at the molecular level.

The paper presents an interesting finding regarding gabapentin treatment in glioblastoma patients. However, without more comprehensive molecular analyses and clear demonstration of target engagement in the tumor tissue, the mechanistic basis for the observed survival benefit remains inadequately supported. These additional analyses would be essential for informing future prospective trials and identifying which patients might benefit most from gabapentin treatment.

Authors' Response:

See response to Comments 3.1 and 3.2.

REVIEWER 4**Comment 4.1:**

Reviewer #4 (Remarks to the Author):

Here's a revised version of your feedback:

Dr. Joshua D. Bernstock (MD, PhD, MPH) and Dr. Shawn Hervey-Jumper (MD) demonstrated a substantial survival benefit associated with gabapentin, an antagonist of TSP-1, following surgical resection in newly diagnosed glioblastoma patients. In this revised analysis, the authors included a total of 1,072 patients, with a discovery cohort of 693 patients and an additional 379 patients from a separate site for external validation. I suggest that the authors revise the subtitle in the results section to clarify these groups as "Discovery and Validation Cohorts."

Authors' Response:

We thank the Reviewer for this comment, fully agree and have updated the manuscript accordingly.

Examples of changes to the text:

“Discovery cohort, Mass General Brigham” (Line 134)

“Validation cohort, University of California, San Francisco” (Line 166)

Comment 4.2:

A key issue arises from the finding that gabapentin acts as an antagonist of TSP-1, yet the authors observed that overall survival was not significantly associated with TSP-1 expression levels at initial resection ($p = 0.09$; Supplementary Figure 2). This result seems contradictory to their hypothesis. I recommend that the authors compare overall survival (OS) and progression-free survival (PFS) among three groups: TSP-1 positive patients treated with gabapentin, TSP-1 positive patients not treated with gabapentin, and TSP-1 negative patients. This analysis could clarify the role of TSP-1 expression in gabapentin’s effect on survival outcomes.

Authors’ Response:

All TSP-1 positive tumors respond to gabapentin, which is the critical/actionable finding of this study.

However, there are several potential explanations for why initial TSP-1 expression did not appear to influence the response to gabapentin. First, the statistical power within these populations may have been insufficient to detect a significant effect. Second, the tumor blocks analyzed may not have been taken from regions of relevant neuronal-glioma interaction (e.g., areas of high functional connectivity), and thus may not accurately represent the overall tumor behavior in a given patient. Future prospective studies, which we have designed, will aim to more precisely delineate the relationship between TSP-1 expression and tumor connectivity at specific sites.

Finally, we would like to highlight that the analysis suggested by the Reviewer would be confounded by the inclusion of TSP-1 negative patients who received gabapentin and is again limited by the number of available patients. Addressing these complexities in future studies will be crucial for further elucidating the role of TSP-1 expression in gabapentin responsiveness.

References

1. (NCBI) NCfBI. TSP1 thrombospondin 1 (human) Gene - Gene ID: 7057. Accessed November 25, 2024, <https://www.ncbi.nlm.nih.gov/gene/7057>

We thank the reviewers for taking the time to review the second revision and for providing thoughtful comments. Reviewers' comments are presented in **bold**, our responses are in **green**, original text in **blue**, and changes to the text in **red**.

REVIEWER 3

Reviewer #3 (Remarks to the Author)

This study by Bernstock et al. investigates an important clinical question regarding the potential therapeutic benefit of gabapentin in glioblastoma patients. The authors present intriguing retrospective data suggesting improved survival in gabapentin-treated patients across two independent cohorts, which represents a potentially significant finding given the limited treatment options for this devastating disease.

Strengths:

- Large patient cohorts (693 discovery + 379 validation patients) from major academic centers**
- Consistent survival benefit observed across independent cohorts**
- Preliminary evidence of target engagement through serum TSP-1 measurements**
- Clear clinical relevance given gabapentin's established safety profile and immediate potential for repurposing**
- Robust statistical methodology including controls for confounding variables**

Limitations:

Lack of comprehensive molecular characterization:

- No methylation array analyses**
- No single-cell or spatial transcriptomics**
- Limited ability to identify molecular predictors of response**
- Insufficient understanding of potential selection biases**

Inadequate demonstration of target engagement:

- Only serum TSP-1 data provided without tissue-level validation**
- No analysis of differential effects on tumor vs microenvironment cells**
- Limited understanding of synaptic density changes and pathway activation**
- Small number of matched primary/recurrent specimens**

Variable treatment parameters:

- Heterogeneous dosing and duration of gabapentin therapy**
- Limited ability to establish dose-response relationships**
- Unclear optimal treatment regimen**

Mechanistic gaps:

- Insufficient correlation between functional connectivity and treatment response
- Limited understanding of which patient subgroups may benefit most
- Unclear relationship between TSP-1 expression and clinical benefit

Recommendations:

While the clinical observations are compelling and potentially impactful, several key analyses would strengthen the manuscript:

- Even with limited tissue availability, targeted molecular analyses of available samples could provide crucial insights into response mechanisms
- More detailed analysis correlating tumor location, functional connectivity, and treatment response
- Better characterization of the relationship between TSP-1 expression and clinical benefit
- Clearer articulation of the limitations and need for prospective validation

Conclusion:

The manuscript presents potentially important clinical findings that warrant serious consideration. However, the lack of molecular and mechanistic characterization limits our ability to fully interpret the results and optimize future clinical applications. While acknowledging the challenges of retrospective analyses, additional molecular studies would significantly strengthen the manuscript's impact and clinical utility.

The central question is whether the clinical significance of the findings outweighs the current mechanistic limitations. Given the urgent need for new therapeutic approaches in glioblastoma, there is merit to publishing these findings to enable further investigation. However, this should be balanced against the risk of prematurely promoting a therapeutic approach without fully understanding its biological basis.

A potential compromise would be to:

More explicitly acknowledge the current limitations

Better define the key questions for prospective investigation

Provide preliminary molecular analyses where possible with available samples

Frame the findings as hypothesis-generating rather than definitive

This would allow the field to build on these observations while maintaining appropriate scientific rigor.

Authors' Response:

We thank the Reviewer for again reviewing our manuscript and for providing such thoughtful feedback throughout this process. In addition, we appreciate and agree with the compromise put forth by the Reviewer.

As such, we have:

1. Expanded the limitations to further acknowledge the points raised by the Reviewer throughout the process, including: the lack of molecular characterization, limitations to demonstration of target engagement, and limitations stemming from the retrospective study design (e.g. dosing details).

2. We agree that future prospective trials, planned by us and others, deserve more attention and these have been further detailed.
3. We are now also more conservative in the conclusions throughout the manuscript given the limitations to these initial pilot data.

Finally, while we cannot add further molecular data, we hope that these changes would adequately address the points raised by the reviewer.

Changes to the text:

“In line with these findings and controlling for critical confounding variables, we demonstrate a **substantial** survival benefit associated with gabapentin (an antagonist of TSP-1) following surgical resection of newly diagnosed glioblastoma.”

~~“However, no studies have yet demonstrated”~~ **“In this work, we investigated if”**

~~“However, larger, randomized, multicenter studies will ultimately be needed to confirm the findings presented and, in doing so, provide more robust evidence for the use of neuromodulators/anticonvulsant medications as adjuvant therapies in GBM, and to identify optimally responsive sub-populations of patients (e.g. based on TSP-1 quantification at initial resection, serum TSP-1, or structural/functional connectivity). Additionally, the absence of stratified molecular subanalyses and the inability to validate tissue-level target engagement further underscores the need for deeper mechanistic investigations.”~~

Additionally, our group is in the final stages of planning a prospective randomized study to assess the impact of gabapentin on overall survival in GBM **with standardized dosing regimens to evaluate dose-response relationships, detailed molecular profiling (e.g. transcriptomics and direct measures of target engagement), and longitudinal follow-up to assess progression-free survival.**

~~This preliminary finding provides further evidence that gabapentin-mediated TSP-1 inhibition (corresponding to reduction of peripheral TSP-1) impairs the proliferation of GBM, which in previous work targeted cell subpopulations within intratumoral regions with high connectivity, thereby reducing peripheral TSP-1.~~

~~We also did not establish a clear relationship between TSP-1 expression level of the tumor and gabapentin response or prognosis in this retrospective study.~~

~~Taken together with prior evidence, this work provides further support for repurposing of gabapentinoids and for TSP-1 inhibition for treatment of GBM, which will be further evaluated in prospective trials.~~

REVIEWER 4

Comment 4.1.

Reviewer #4 (Remarks to the Author)

This is the third version of the manuscript. I commend the authors for their dedication and hard work in refining the study. Below, I outline my key comments:

A central issue arises from the finding that gabapentin acts as an antagonist of TSP-1, yet the authors report that overall survival (OS) is not significantly associated with TSP-1 expression levels at initial resection ($p = 0.09$; Supplementary Figure 2). This result appears to contradict the authors' hypothesis. I recommend that the authors conduct a more detailed analysis by comparing OS and progression-free survival (PFS) across three distinct groups: (1) TSP-1 positive patients treated with gabapentin, (2) TSP-1 positive patients not treated with gabapentin, and (3) TSP-1 negative patients. This subgroup analysis may provide greater insight into the role of TSP-1 expression in mediating gabapentin's effects on survival outcomes. Despite reviewing the authors' response, I remain unclear on this issue.

Authors' Response:

We thank the reviewer for this point, we have provided the suggested analysis below (in patients stained for TSP-1 expression).

As pointed out in our prior response, TSP-1 negative patients include both gabapentin and non-gabapentin patients and does not serve as an adequate control arm. However, we have added the analysis to the supplemental data. With respect to PFS, these data were not readily available to our group, but we acknowledge it to be an important point, and have included it as a point to consider the future trials section of the manuscript (see response to **REVIEWER 3**).

Changes to the text:

“However, larger, randomized, multicenter studies will ultimately be needed to confirm the findings presented and, in doing so, provide more robust evidence for the use of neuromodulators/anticonvulsant medications as adjuvant therapies in GBM, and to identify optimally responsive sub-populations of patients (e.g. based on TSP-1 quantification at initial resection, serum TSP-1, or structural/functional connectivity). Additionally, the absence of stratified molecular subanalyses and the inability to validate tissue-level target engagement further underscores the need for deeper mechanistic investigations.”

Comment 4.2.

Regarding the second point, the authors acknowledge the concern in principle and have outlined plans to design future prospective studies to address these goals. However, further clarification on how these studies will be structured and how they will address the current limitations would be helpful.

Authors' Response:

We thank the Reviewer for this important point. We agree that future prospective trials deserve a more detailed description, a third revision of the manuscript would include

suggestions to further assess molecular patterns, dose response patterns and PFS. We have made the following changes to the text.

Changes to the text:

“Additionally, our group is in the final stages of planning a prospective randomized study to assess the impact of gabapentin on overall survival in GBM with standardized dosing regimens to evaluate dose-response relationships, detailed molecular profiling (e.g. transcriptomics and direct measures of target engagement), and longitudinal follow-up to assess progression-free survival.”